Tropical peatlands and their conservation are important in the context of COVID-19 and potential future (zoonotic) disease pandemics

http://orcid.org/0000-0002-0729-8407 Harrison Mark E. 1 2 3 m.e.harrison@exeter.ac.uk
http://orcid.org/0000-0001-5030-6962 Wijedasa Lahiru S. 4 5
http://orcid.org/0000-0003-3198-6311 Cole Lydia E.S. 6
http://orcid.org/0000-0002-9180-3356 Cheyne Susan M. 2 7 8
Choiruzzad Shofwan Al Banna 9 10
http://orcid.org/0000-0001-7518-8181 Chua Liana 11
http://orcid.org/0000-0002-1871-6360 Dargie Greta C. 12
http://orcid.org/0000-0001-5622-5127 Ewango Corneille E.N. 13
http://orcid.org/0000-0003-2314-590X Honorio Coronado Euridice N. 14
http://orcid.org/0000-0003-3351-9987 Ifo Suspense A. 15
http://orcid.org/0000-0003-2371-7795 Imron Muhammad Ali 16
Kopansky Dianna 17
Lestarisa Trilianty 18 19
O’Reilly Patrick J. 3
Van Offelen Julie 20
Refisch Johannes 21
Roucoux Katherine 6
Sugardjito Jito 22 23
Thornton Sara A. 2 3
Upton Caroline 3
Page Susan 2 3
1 Centre for Ecology and Conservation, College of Life and Environmental Sciences, University of Exeter , Penryn , UK
2 Borneo Nature Foundation International , Penryn , UK
3 School of Geography, Geology and the Environment, University of Leicester , Leicester , UK
4 Integrated Tropical Peatland Research Program (INTPREP), Environmental Research Institute, National University of Singapore , Singapore , Singapore
5 ConservationLinks Pvt Ltd , Singapore , Singapore
6 School of Geography and Sustainable Development, University of St. Andrews , St. Andrews , UK
7 Humanities and Social Sciences, Oxford Brookes University , Oxford , UK
8 IUCN SSC PSG Section on Small Apes , Oxford , UK
9 Department of International Relations, Universitas Indonesia , Depok , Indonesia
10 ASEAN Studies Center, Universitas Indonesia , Depok , Indonesia
11 Department of Social and Political Sciences, Brunel University , London , UK
12 School of Geography, University of Leeds , Leeds , UK
13 Faculty of Renewable Natural Resources Management/Faculty of Sciences, University of Kisangani , Kisangani , DR Congo
14 Instituto de Investigaciones de la Amazonía Peruana , Iquitos , Perú
15 Laboratoire de Géomatique et d’Ecologie Tropicale Appliquée, Département des Sciences et Vie de la Terre, Ecole Normale Supérieure, Université Marien Ngouabi , Brazzaville , Republic of Congo
16 Faculty of Forestry, Universitas Gadjah Mada , Yogyakarta , Indonesia
17 Global Peatlands Initiative, Ecosystems Division, United Nations Environment Programme , Nairobi , Kenya
18 Faculty of Medicine, Palangka Raya University , Palangka Raya, Kalteng , Indonesia
19 Doctoral Program of Public Health, Airlangga University , Surabaya , Indonesia
20 Independent Consultant , Nairobi , Kenya
21 Great Apes Survival Partnership, United Nations Environment Programme , Nairobi , Kenya
22 Centre for Sustainable Energy and Resources Management, Universitas Nasional , Jakarta , Indonesia
23 Faculty of Biology, Universitas Nasional , Jakarta , Indonesia
Roberts David
Electronic publication date: 2020 Nov 17
Publication date: 2020
Volume: 8
Electronic Location ID: e10283
Received 2020 Jun 24; Accepted 2020 Oct 9
Copyright: © 2020 Harrison et al.
Copyright year: 2020
Copyright holder: Harrison et al.
License: This is an open access article distributed under the terms of the Creative Commons Attribution License, which permits unrestricted use, distribution, reproduction and adaptation in any medium and for any purpose provided that it is properly attributed. For attribution, the original author(s), title, publication source (PeerJ) and either DOI or URL of the article must be cited.
License URL: https://creativecommons.org/licenses/by/4.0/

Keywords: Amazon, Congo basin, Conservation, Coronavirus, Emerging infectious disease (EID), Indonesia, SARS-CoV-2, Southeast Asia, Wildlife harvesting, Zoonoses

Funding: NERC-GCRF NE/T010401/1 Borneo Nature Foundation International The NERC-GCRF (grant no.: NE/T010401/1) and Borneo Nature Foundation International provided financial contributions towards this work. There was no additional external funding received for this study. The funders had no role in study design, data collection and analysis, decision to publish, or preparation of the manuscript.

==============================
The COVID-19 pandemic has caused global disruption, with the emergence of this and other pandemics having been linked to habitat encroachment and/or wildlife exploitation. High impacts of COVID-19 are apparent in some countries with large tropical peatland areas, some of which are relatively poorly resourced to tackle disease pandemics. Despite this, no previous investigation has considered tropical peatlands in the context of emerging infectious diseases (EIDs). Here, we review: (i) the potential for future EIDs arising from tropical peatlands; (ii) potential threats to tropical peatland conservation and local communities from COVID-19; and (iii) potential steps to help mitigate these risks. We find that high biodiversity in tropical peat-swamp forests, including presence of many potential vertebrate and invertebrate vectors, combined, in places, with high levels of habitat disruption and wildlife harvesting represent suitable conditions for potential zoonotic EID (re-)emergence. Although impossible to predict precisely, we identify numerous potential threats to tropical peatland conservation and local communities from the COVID-19 pandemic. This includes impacts on public health, with the potential for haze pollution from peatland fires to increase COVID-19 susceptibility a noted concern; and on local economies, livelihoods and food security, where impacts will likely be greater in remote communities with limited/no medical facilities that depend heavily on external trade. Research, training, education, conservation and restoration activities are also being affected, particularly those involving physical groupings and international travel, some of which may result in increased habitat encroachment, wildlife harvesting or fire, and may therefore precipitate longer-term negative impacts, including those relating to disease pandemics. We conclude that sustainable management of tropical peatlands and their wildlife is important for mitigating impacts of the COVID-19 pandemic, and reducing the potential for future zoonotic EID emergence and severity, thus strengthening arguments for their conservation and restoration. To support this, we list seven specific recommendations relating to sustainable management of tropical peatlands in the context of COVID-19/disease pandemics, plus mitigating the current impacts of COVID-19 and reducing potential future zoonotic EID risk in these localities. Our discussion and many of the issues raised should also be relevant for non-tropical peatland areas and in relation to other (pandemic-related) sudden socio-economic shocks that may occur in future.

Introduction

The COVID-19 pandemic has caused unprecedented global disruption, at the time of writing infecting over 50 million and killing over a million people across the globe (Dong, Du & Gardner, 2020). These health impacts, plus lockdowns and other measures to control the pandemic, have resulted in reduced economic activity and job losses, leading to potentially the worst global recession since the Great Depression (IMF, 2020). While these global economic and social disruptions have had a positive, albeit likely temporary, impact on global carbon emissions (Le Quéré et al., 2020), negative outcomes are widely expected for biodiversity conservation, research (Corlett et al., 2020; Evans et al., 2020; Hockings et al., 2020; Lindsey et al., 2020) and indigenous communities (IUCN, 2020; UN/DESA, 2020; UN/EMRIP, 2020).

COVID-19 cases and deaths have been recorded for most tropical countries, with high numbers in some (Dong, Du & Gardner, 2020), and with testing shortfalls likely partially accounting for low reported case numbers for some other tropical nations, particularly in Africa (Ditekemena, 2020; Nordling, 2020). Many tropical nations are low- or middle-income countries, with weaker health systems and fewer resources to tackle the pandemic, generating further concerns. Some of the countries with high infection and mortality rates also have large remaining areas of tropical peatland, including Brazil, Peru, Ecuador and Indonesia (Dong, Du & Gardner, 2020; Page, Rieley & Banks, 2011). While currently reporting relatively few cases, the Congo Basin contains the world’s largest tropical peatland area (Dargie et al., 2017), is among the most poorly resourced to tackle disease pandemics in general (Oppenheim et al., 2019) and COVID-19 in particular (World Bank, 2020a), and the Democratic Republic of Congo (DRC) is projected to suffer a substantial number of cases and deaths (Cabore et al., 2020).

The importance of healthy tropical peatlands for carbon storage and emission mitigation, conserving biodiversity and providing ecosystem services for local communities is increasingly recognised (Baker et al., 2020; Crump, 2017; Dargie et al., 2017; Dommain et al., 2016; Hooijer et al., 2009; Husson et al., 2018; Page, Rieley & Banks, 2011; Posa, Wijedasa & Corlett, 2011), but to our knowledge no published study has specifically considered tropical peatlands and their inhabitants in the context of emerging infectious disease (EID), although some infectious disease studies have been conducted in tropical peatland areas (Vittor et al., 2006). Addressing this gap is important because ongoing land-use change is reducing tropical peat-swamp forest (TPSF) coverage, while bringing an increasing number of human communities in close contact with peatlands (Field, Van der Werf & Shen, 2009; Parish et al., 2008) and thus their biodiversity, although to date this has occurred far less in South American and African peatlands, compared to South-east Asia (Dargie et al., 2019; Roucoux et al., 2017). Such an assessment is made more urgent considering the ongoing COVID-19 pandemic and its potential impacts, and the global trend for increased EID event incidence (Jones et al., 2008). It is also relevant for understanding and reducing the potential for emergence of, and impacts arising from, any future EIDs in tropical peatland nations.

Our goals in this paper are thus to present a preliminary synthesis of: (i) the potential for future EID (re-)emergence from tropical peatlands; (ii) potential threats to tropical peatland conservation and local communities from the current COVID-19 pandemic; and (iii) potential steps to help mitigate these risks. These goals cover a very broad range of potential topic areas, ranging from local livelihoods and food security, to habitat conservation efforts and scientific research, among others. A summary illustrating some key features relating to (i) and (ii) is provided in Fig. 1. Although focused on tropical peatlands, many of the issues discussed and concerns raised will also be relevant to non-peatland areas in the tropics. This synthesis should thus be of interest to researchers, conservation/restoration/community project proponents, land managers and policy makers in the tropics, especially in but not restricted to peatland areas. While framed in the context of COVID-19, it is also pertinent to note that our discussion and many of the issues raised are in reality not uniquely linked to COVID-19, but rather relate more generally to (pandemic-related) sudden socio-economic shocks (e.g. economic recessions, border closures due to other causes, or extreme events related to climate change) that may occur in future.

Figure 1 Summary illustration of key points and linkages regarding the potential for future EID emergence from tropical peatlands, and potential threats to tropical peatland conservation and local communities from the current COVID-19 pandemic.

Box outline and fill colours indicate contents corresponding to labels of the three circles. Dashes indicate pre-existing factors or factors arising from the COVID-19 pandemic that impact upon solid boxes. Shaded boxes indicate potential major ultimate impacts, which as implied by the three circles are closely linked and thus potentially mutually reinforcing. See text for further details, explanations and discussion of additional points and linkages omitted here for reasons of space. Photo credits: Sara Thornon (tropical peat-swamp forest), Susan Page (peatland communities & livelihoods) and Philippa Steinberg for the Innovative Genomics Institute (EIDs/COVID-19: Steinberg, 2020).

Survey Methodology

We conducted a scoping review of relevant literature in relation to the above goals. First, we conducted a structured search of scientific databases using search terms related to these topics (Table S1). This approach reduces potential for bias in awareness among our author team, but yielded very few potentially relevant studies (total n = 6, Table S2), owing to a lack of past studies concerning EIDs, including COVID-19, in the context of tropical peatlands. These results alone were insufficient to draw meaningful conclusions relating to any one of our goals, whereas conducting similarly structured database searches around every potential topic of relevance to EIDs/COVID-19 in the context of tropical peatlands would have been impractical, given the huge variety of potentially relevant topics relating to our goals, large number of countries containing tropical peatlands and the fact that these countries are not entirely covered in tropical peatlands, thus limiting potential to use individual tropical peatland nation names as search criteria. Consequently, we also conducted a less formally structured literature review, drawing on the subject knowledge, awareness of formal and informal literature sources, personal experiences and networks of our author team. In line with our scoping aims and very broad line of questioning, this did not employ strict inclusion and exclusion criteria, asides from excluding studies of no conceivable relevance to EIDs/COVID-19 and tropical peatlands. Such an approach allows us to draw relevant information from the many studies and reports of potential relevance that we were aware of that do not specifically concern EIDs/COVID-19 in the context of tropical peatlands and that did not therefore appear in our structured searches. The remainder of this manuscript is therefore focused on this much more informative less formally structured review, though findings from the structured search are integrated in relevant sections of the text.

To help ensure comprehensive coverage of the literature, and minimise geographic and subject bias in this review, our author team was developed to include natural and social scientists with substantial direct experience of, and familiarity with, the literature in relation to tropical peatland research in South-east Asia, Africa and South America (see, for example past reviews: Dargie et al., 2019; Harrison et al., 2020; Page & Hooijer, 2016; Roucoux et al., 2017). While we attempt to provide context across all four continents on which tropical peatlands are found, we acknowledge some bias towards South-east Asia, for which a far greater volume of information is available. Where possible we draw upon peer-reviewed and other highly reputable sources (e.g. UN reports), but owing to the COVID-19 pandemic’s recent emergence and consequent paucity of such literature relating specifically to it in the context of the issues considered, we also draw upon pre-prints and media reports where peer-reviewed sources are unavailable or just provide general (rather than COVID-19 specific) support for a statement. We attempt to indicate such cases clearly and verify these from multiple sources, while noting that this reflects the uncertainty and rapid evolution of the pandemic and associated public debate.

Are tropical peatlands a potential source habitat for disease pandemics?

Most EID events are dominated by zoonoses (60.3%), with the majority of these (71.8%) originating in wildlife, including Acquired Immunodeficiency Syndrome (AIDS), Severe Acute Respiratory virus (SARS), Middle East Respiratory Syndrome (MERS) and Ebola virus (Jones et al., 2008), plus the novel COVID-2019, an ongoing global pandemic as we write this paper (Li et al., 2020). In Africa, for example, 25 types of parasites, nine main types of viruses and eight types of bacteria have been reported as present in wild meat and communicable to humans (Van Vliet et al., 2017). The joint-first reported case of Ebola in 1976 is from a peatland area (Yambuku, DRC: Muyembe-Tamfum et al., 2012), as is the most recent outbreak in May 2020 (Mbandaka, DRC: WHO, 2020c; Fig. 2), and the cradle of the HIV/AIDS pandemic is believed to be around Kinshasa, DRC, another area with extensive peatlands (Sharp & Hahn, 2011; Worobey et al., 2008). The risk of zoonotic EID emergence is positively correlated with high human population density and wildlife host richness (Allen et al., 2017; Jones et al., 2008); wild animal harvesting and/or movement of animals or body parts, leading to increased contact between wildlife vectors and humans (Bengis et al., 2004; Johnson et al., 2020); biodiversity loss (Keesing et al., 2010); spread of non-indigenous vectors and pathogens; plus habitat encroachment, fragmentation and alteration (Allen et al., 2017; Johnson et al., 2020; Pongsiri et al., 2009).

Figure 2 Map of tropical and subtropical peatlands (A) following Leifeld & Menichetti (2018), with close-ups of the Congo Basin (B) and Riau province, Sumatra, Indonesia (C).

In (B) and (C) the satellite image is a false colour composite of Landsat 7 and 8 imagery from 2016 to 2017 created in GoogleEarth Engine. Human population density per square kilometre (Gaughan et al., 2013; Linard et al., 2012; WorldPop, 2020) is overlaid with colours ranging from pink, with a density of one person per square kilometre, to greater than 50 in red. Peatland boundaries are indicated by yellow lines and purple indicates treeless areas. Most of the Congo Basin peatlands have low population density, but there is Mbandaka city in the centre (red), plus a growing population in the periphery on the northeast and east. In Riau, there are large population centres, plus growing populations and plantations directly adjacent to the peatlands. Map data: Google, USGS; Gaughan et al. (2013), Linard et al. (2012), Leifeld & Menichetti (2018) and WorldPop (2020).

The natural habitat of tropical peatlands, tropical peat-swamp forest (TPSF), possesses a rich fauna and flora, including numerous vertebrate taxa known to represent zoonotic EID risk, such as bats, rodents, pangolins and primates (Husson et al., 2018; Inogwabini et al., 2012; Posa, Wijedasa & Corlett, 2011). Indeed, previous studies on primates and small mammals in South-east Asian TPSF areas have recorded numerous parasite species found in humans and that are of medical concern (Hilser, 2011; Hilser, Ehlers Smith & Ehlers Smith, 2014; Madinah et al., 2011, 2014; Nurcahyo, Konstanzová & Foitová, 2017), and surveys of bats from TPSF areas in Peru have detected high rates of Bartonella bacteria infection (Bai et al., 2012), suggesting potential for disease transmission from humans to wildlife and zoonotic transmission from wildlife to humans in TPSF areas. Studies conducted in non-TPSF areas on species that are also found in TPSF support this conclusion (e.g. chimpanzee Pan troglodytes deaths from human paramyxoviruses in Ivory Coast: Köndgen et al., 2008).

Tropical peat-swamp forest conversion, plus fire and wildlife harvesting bring more people into closer contact with peatland biodiversity. In South-east Asia, degradation, fragmentation and conversion of TPSF to agriculture has been particularly widespread, with the area of peatland in Malaysia, Sumatra and Kalimantan covered by TPSF declining from 76% (11.9 Mha) in 1990 to 29% (4.6 Mha) in 2015, with a concomitant increase from 11% (1.7 Mha) to 50% (7.8Mha) of the area covered by agriculture over the same time period (Miettinen, Shi & Liew, 2016). This near doubling of agricultural extent has been driven by small-scale farmers (43–44%), plus industrial oil palm (39%) and paper pulp expansion (11–26%) (Miettinen, Shi & Liew, 2016; Wijedasa et al., 2018). Further forest habitat fragmentation is expected if all planned road and rail infrastructure development projects on Kalimantan proceed as proposed, with a projected reduction in landscape connectivity from 89% to 55% (Alamgir et al., 2019).

Such threats are not limited to South-east Asia. For example, although there is currently limited encroachment into the TPSFs of the Peruvian Amazon from industrial agriculture and infrastructure development (Lilleskov et al., 2019), these threats are present and considerable (Baker et al., 2020; Roucoux et al., 2017), and could increase the likelihood of EID emergence from the Amazon in the near future (Ellwanger et al., 2020). Likewise, projected mining permits, gas and oil exploration, timber and palm oil concessions, associated road construction and changing rainfall patterns due to global warming pose potential risks to the relatively undisturbed TPSF of the Congo Basin and its wildlife (Dargie et al., 2019; Haensler, Saeed & Jacob, 2013; Miles et al., 2017; Wich et al., 2014). In the Republic of Congo (RoC), health authority reports claim that several recent malaria infections appeared in swampy forests for the first time following opening of logging trails (Rédaction, 2020), highlighting the potential for increased EID risk should these peatlands experience greater future encroachment.

Patterns of human-wildlife contact and wild meat hunting in tropical peatlands provide further potential for disease spill-over events (during which a pathogen from one species moves into another) from wildlife to humans to occur. Human population density in tropical peatland areas is typically not high (e.g. Peru: Lilleskov et al., 2019; Congo Basin: Dargie et al., 2019), but some large population centres are found close to TPSF (e.g. Jambi and Palangka Raya, Indonesia; North and South Selangor, Malaysia; Iquitos, Peru; Fig. 2), and certain human activities increase contact between people and potential animal vectors in TPSF. Wildlife harvesting for consumption and trade is common in tropical forest nations (Fa, Currie & Meeuwig, 2003; Nielsen et al., 2018), including in TPSF areas. For example, in Central Kalimantan, Indonesia, Pteropus vampyrus (Linn.) fruit bats are captured in TPSF areas and transported to local markets for sale as wild meat (Harrison et al., 2011). High contact levels between bats, hunters and vendors occur, with hunters and vendors frequently bitten and most bites drawing blood, raising concerns regarding potential zoonotic disease transmission (Harrison et al., 2011), though recent anecdotal observations suggest that local demand and trade has decreased as a likely result of the pandemic (R. Dwi et al., 2020, personal communication). Other species are commonly harvested for commercial trade in tropical peatland nations, often for sale in dense population areas, including turtles (Schoppe, 2009) and primates (Nijman et al., 2015) in Indonesia; plus pangolins and numerous other species from across South-east Asia (Nijman, 2010). Similar unsustainable hunting has been reported in the Congo Basin (Fa, Currie & Meeuwig, 2003; Poulsen et al., 2009). In the Peruvian Amazon TPSF, tapirs (Tapirus terrestris, Linn.), primates, rodents and other mammals are commonly hunted by local communities (Schulz et al., 2019a). Although there is some wildlife export to local market centres (e.g. Iquitos: Bodmer & Lozano, 2001; Mayor et al., 2019), most is consumed at a household level and constitutes an important protein source (Schulz et al., 2019a), providing higher potential to limit wildlife export from peatland areas than may be the case in South-east Asia and Africa.

High densities of domestic and semi-wild animals reared on peatlands could also serve as a direct or indirect zoonotic EID vector to humans. For instance, in Indonesia, over 1.8 million chickens are kept in the predominantly peatland municipality of Palangka Raya (Statistics of Palangka Raya Municipality, 2018), while large numbers of naturally cave-roosting edible-nest swiftlets (mostly Aerodramus spp.) are reared in special buildings in many peatland areas, with most nests exported to China (Husson et al., 2018; Koon & Cranbrook, 2014; Thorburn, 2014).

In summary, the combination of high native faunal diversity, habitat encroachment and fragmentation, plus trade in native and non-native fauna in many tropical peatland areas appears to represent a suitable set of conditions under which zoonotic EIDs could potentially (re-)emerge in future. Given that TPSFs represent the largest remaining blocks of lowland forest in some areas (e.g. lowland Borneo: Wijedasa et al., 2018; Congo Basin: Dargie et al., 2017; parts of the Amazon: Roucoux et al., 2017), attention should be paid to conserving and sustainably managing these TPSFs and their wildlife to reduce the likelihood of potential future zoonotic EID pandemics arising.

What are the potential immediate impacts of the COVID-19 pandemic in tropical peatland areas?

The impacts of the COVID-19 pandemic will depend heavily on its length and severity, and evolving government and societal responses, which will vary between and potentially even within tropical peatland nations. This unpredictability notwithstanding, we nevertheless outline some areas of potential concern relating to tropical peatlands specifically, while referring readers to the more generic issues raised by Corlett et al. (2020), Evans et al. (2020) and Hockings et al. (2020), many of which will apply to tropical peatlands as much as to other habitats, and to any potential future pandemics causing similar levels of socio-economic disruptions to COVID-19. It is also pertinent to note that these issues occur on top of pre-existing challenges for tropical peatland conservation and sustainable management (see Dargie et al., 2019; Harrison et al., 2020; Roucoux et al., 2017 for reviews).

Public health and potential combined impacts from haze pollution

Disadvantaged populations are expected to be disproportionately affected by pandemics, further exacerbating existing social and economic inequalities (Lee, Rogers & Braunack-Mayer, 2008; WHO, 2009). With the exception of Reunion, Brunei, Puerto Rico and Australia, which contain only small peatland areas, all tropical peatland nations listed by Page, Rieley & Banks (2011) are classified as low or middle income (OECD, 2020), with many also being considered relatively under-prepared to cope with disease pandemics (Oppenheim et al., 2019). Tropical peatland communities are often relatively remote and (agricultural) conditions marginal, with lower access to public health and other services, no or poor medical insurance, fewer formal employment opportunities and higher poverty rates than non-peatland areas (e.g. Kalimantan: Medrilzam et al., 2014; Thornton, 2017; Van Beukering et al., 2008; Wiseman et al., 2018; DRC: C. Ewango & G. Dargie, 2020, personal observation; Peru: E. Honorio, 2019, personal observation; Fig. 3). The rural populations of several tropical peatland nations have disproportionate numbers of people with underlying health conditions and/or malnutrition (Kandala et al., 2011; Nair, Wares & Sahu, 2010), and many do not have access to formal healthcare, or the running water, good sanitation and hygiene systems required to implement the recommended WASH approach to COVID-19 (WHO & UNICEF, 2020). For example in Central Kalimantan, less than 40% of people have access to improved sanitation (WHO, 2017) and the province tends to perform poorly in healthcare provision evaluations (Suparmi et al., 2018; Wiseman et al., 2018). Disseminating COVID-19 health guidance information will likely also be more difficult in rural tropical peatland areas with poor communications infrastructure, further reducing the probability that risk reduction behaviours will be followed. It is therefore possible that the health impacts of the COVID-19 pandemic may be relatively severe and/or prolonged in tropical peatland nations.

Figure 3 A remote tropical peatland community in Buenos Aires (A), within the Pacaya Samiria National Reserve, Peruvian Amazon, which is accessible only by boat.

People living here and in neighbouring communities rely heavily on resources extracted from the surrounding peat-forming Mauritia flexuosa palm swamps. Urarina indigenous groups living in peat-rich areas harvest palm leaves from which to make textiles (B); important both practically and culturally for these isolated communities. The palms also offer plentiful food for wild fauna and thus the palm swamps in which they grow are important hunting spaces for people, providing bushmeat in locations far from the nearest market. Photo credits: Lydia Cole.

While their remoteness and low population densities may reduce the potential for COVID-19 to reach and spread between some rural tropical peatland communities, evidence to support this supposition is limited, since many of these communities—like many others on resource frontiers—are deeply embedded in market relations (Li, 2014; Medrilzam et al., 2014) and local to international value chains bring them into regular contact with outsiders (Dove, 2011; Schreer, 2016). Even among those communities living relatively autonomously, there are still levels of contact with non-community actors, including NGOs, researchers, public agencies and commercial organisations. Indeed, media reports from the Americas and Africa indicate that COVID-19 has already spread into remote indigenous communities (Brito, 2020; Wallace, 2020). In DRC, this comes on top of existing Ebola and measles epidemics (Blomfield, 2020), and in Iquitos, Peru, on top of dengue fever and leptospirosis outbreaks (Collyns, 2020b). Once introduced, the possibility of COVID-19 cases going undetected and unreported, owing to a lack of testing, limited awareness plus an element of fatalism arising from the commonness of disease in such communities (e.g. Borneo: L. Chua, 2003–2020, personal observation), combined with the important role that communal activities and high levels of social interaction play in many peatland communities, raises the risk of infection spreading. The more limited public health resources in rural tropical peatland areas means that the health impacts arising should an outbreak occur are potentially more serious than in less remote and more affluent communities. Furthermore, development and medical assistance in tropical peatland areas may be temporarily halted due to financial and organisational challenges (e.g. mobile medical teams in Peruvian Amazon: Vine Trust, 2020). These considerations place an onus on businesses, governments, NGOs and researchers working with such communities to take measures to reduce the chances of introducing and spreading the disease between communities.

Given that elderly people are more vulnerable to COVID-19 (Zhou et al., 2020), one impact of outbreaks in tropical peatland areas may be a loss of traditional local knowledge regarding these ecosystems. Another important knock-on impact highlighted by the WHO is potential for disrupted responses to other major public health issues that may risk reversing gains made against these. This includes malaria and dengue fever, which are endemic in many tropical peatland areas and exhibit several symptoms similar to COVID-19 (PAHO, 2020; WHO, 2020d); and immunisations for diseases such as diphtheria, measles and polio (WHO, 2020a). In addition, co-infection of COVID-19 with other diseases and increased COVID-19 mortality rates in co-infected patients has been reported (Lansbury et al., 2020; Zhou et al., 2020), as has a case of co-infection of COVID-19 and Plasmodium vivax malaria (Sardar et al., 2020; following hospitalisation and discharge, the patient in this case tested negative for both diseases). Important questions therefore exist regarding the potential for (increased impacts from) COVID-19 co-infection in tropical peatland areas, especially for the most vulnerable members of these communities. Measures to tackle these pre-existing diseases must therefore continue to remain a priority in tropical peatland areas.

Tropical peatland degradation and drainage increase fire risk. In Indonesia, peatland fires and their associated haze (smoke pollution) now occur almost annually (Gaveau et al., 2014; Page & Hooijer, 2016; Fig. 4), leading to high carbon emissions, forest and biodiversity losses, and major public health impacts from inhalation of the toxic haze (see Harrison et al., 2020; Page & Hooijer, 2016; Uda, Hein & Atmoko, 2019 for reviews). The haze contains high small particulate concentrations and several toxic compounds, including CO2, CO, CH4, NH3, HCN, NOX, OCS and HCl (Stockwell et al., 2016). Haze exposure during the prenatal period has been linked to decreased adult height attainment (Tan-Soo & Pattanayak, 2019), and short-term exposure during the severe 2015 fires is estimated to have caused 100,300 or more premature mortalities in Equatorial Asia (Koplitz et al., 2016; see also Crippa et al., 2016). This is important in light of recent reports that increased air pollution may elevate COVID-19 case numbers, hospital admissions and mortality (Cole, Ozgen & Strobl, 2020; Conticini, Frediani & Caro, 2020; Ogen, 2020; Wu et al., 2020), and has led to concerns being raised by both regional think tanks (Gan et al., 2020) and relevant experts in media reports regarding peatland fires and COVID-19 (Jong, 2020b; Listiyorini, 2020; Varkkey, 2020), though under COVID-19 lockdown conditions such impacts may be at least partially mitigated by general shutdowns of anthropogenic activities (Kanniah et al., 2020). In particular, in a pre-print article, Cole, Ozgen & Strobl (2020) and Wu et al. (2020) report that an increase in PM2.5 (small particulate matter) of just 1 μg/m3 is associated with a 8–16.6% increase in COVID-19 death rate, whereas in Central Kalimantan, PM2.5 levels have been reported to exceed 1,500 μg/m3 during severe fire periods (Atwood et al., 2016), and average mean exposures between 2011 and 2015 have been estimated at 26 μg/m3, over double the recommended WHO exposure limit (Uda, Hein & Atmoko, 2019). Similar findings were reported in relation to the impacts of air pollution on fatalities from the earlier SARS epidemic in China (Cui et al., 2003), suggesting that high levels of air pollution may increase vulnerability of populations exposed to haze from peatland fires to future pandemics. Observations that SARS-CoV-2 RNA can be present on particulate matter (Setti et al., 2020), and suggestions that particulates, such as PM2.5, are able to penetrate deep inside the lungs and remain in the air for long periods of time (Frontera et al., 2020) lead to further concern that haze pollution from peat fires may increase COVID-19 transmission. Finally, some symptoms of haze exposure are also similar to those of COVID-19 (e.g. dry cough, weakness), which may lead to complications with regards to COVID-19 testing and case identification. While extensive tropical peatland fires are currently mainly limited to Indonesia (Dargie et al., 2019; Lilleskov et al., 2019), increasing pressure to develop African and South American peatlands could elevate their fire risks if preventative measures are not implemented (Roucoux et al., 2017), with consequent potential impacts on the susceptibility of their populations to future respiratory EID pandemics. Continuing and amplifying measures to avoid and control fires in tropical peatlands (Page & Hooijer, 2016; Dohong, Abdul Aziz & Dargusch, 2018; Wijedasa et al., 2018; Harrison et al., 2020) is therefore of heightened importance.

Figure 4 Peatland fire encroaching into forest (A) and local fishers working under thick haze conditions from peatland fires (B) in Central Kalimantan, Indonesia.

Note in (A) the immediate fire damage to the forest on the top side, and older fire damage on the bottom side, of the river. Photo credits: (A) Markurius Sera/Borneo Nature Foundation Indonesia and (B) Suzanne Turnock/Borneo Nature Foundation Indonesia.

Economy and livelihoods

The COVID-19 pandemic and associated response measures are likely to have deep and long-lasting adverse global and local economic impacts, with the worst case scenario being that poverty alleviation achieved in recent decades might be reversed, earnings reduced and our ability to meet the UN Sustainable Development Goal of ending poverty by 2030 compromised (ASEAN, 2020; Lucas, 2020; Sumner, Hoy & Ortiz-Juarez, 2020; World Bank, 2020b). Although tropical peatland communities are less likely to face the types of direct interruptions to livelihood resulting from strict lockdowns, adverse economic impacts linked to the pandemic may nevertheless have important consequences for them. Indeed, low incomes in many tropical peatland households, and weak social and food security safety nets in many tropical peatland countries (FAO, IFAD, UNICEF, WFP & WHO, 2019), may make them particularly vulnerable. Anticipating the nature and the severity of these effects is complicated by the diversity of peatland communities and their livelihood heterogeneity (Jelsma et al., 2017). In Indonesia, for example economic activities include self-employment in commodity tree crop, short-term vegetable crop and livestock (especially chicken) production, swiftlet nest farming, hunting, fishing, gathering non-timber forest products, logging, artisanal mining, employment in construction and other industries, trading, remittances from family members employed elsewhere, and employment by government services and NGOs (Schreer, 2016; Thornton, 2017). The intensity of direct economic impacts arising from the COVID-19 pandemic are likely to be linked to the way communities are integrated into wider trade and resource allocation networks. Here we consider some of these potential impacts.

The anticipated global recession is expected to have a generally negative impact on agricultural prices, with prevalent low incomes in peatland areas likely to amplify the impact of any price falls. Indeed, the UN and IUCN have flagged the expected negative impact of the COVID-19 pandemic on the livelihoods of remote indigenous groups and called for special measures to restore and support traditional indigenous economies (IUCN, 2020; UN/DESA, 2020; UN/EMRIP, 2020). Tropical peatland communities are commonly not food self-sufficient and tend to be close to poverty at the best of times (Wildayana & Armanto, 2018). Global food shortages due to COVID-19 may have implications for the price of imported food, increasing pressure on these communities and potentially forcing some into poverty (as feared more generally by key international institutions: Sánchez-Páramo, 2020; World Food Programme, 2020). Small-scale settler communities cultivating short-lifecycle food crops and staples serving local markets are likely to be less adversely affected, because demand for their output is relatively inelastic and close market proximity means they are better placed to respond to changes in local demand by rapidly switching production. To date, prices of such crops are indeed remaining relatively stable (Lucas, 2020), although some increases are anticipated (Amanta & Aprilianti, 2020). For such producers, the greater problem may be changes in the price of agricultural inputs (e.g. fertilisers) and other imported goods.

Producers of commodities such as palm oil, rubber and beef on tropical peatlands are likely to experience more severe adverse impacts due to falling prices resulting from changes in international demand. For example the value of palm oil fell by over 20% from December 2019 to April 2020 (Index Mundi, 2020), and declining palm oil export volumes and domestic consumption have already been noted in Indonesia (Sarkar, 2020), leading to reported concerns regarding lower demand and falling prices (Gan et al., 2020). Any sustained fall in palm oil price is likely to more severely impact peatland-based production, which is less productive than cultivation on mineral soil, and small-scale producers, for whom yields are typically lower and who will have less resources to sustain them through difficult periods (Euler et al., 2016; Sumarga et al., 2016). These problems are exacerbated by the long-term commitment and substantial investments that these crops require, which restrict the capacity of farmers to rapidly change to other crops, thus limiting their resilience. Prolonged low prices, combined with long-standing issues such as land titling, are thus likely to lead to increased hardship among these small-scale commodity producers, with previous commodity price falls being linked to increased poverty, mental health issues and suicide among small scale producers (Tyson, Varkkey & Choiruzzad, 2018). Conversely, decisions of better resourced, larger plantation companies regarding expansion or contraction of operations will be more heavily influenced by predictions of the longer-term impact of the COVID-19 pandemic, rather than immediate price changes. Current thinking in the palm oil industry appears to be that, while the short-term impact may be significant, longer-term effects are uncertain, given the nature of the product and its market (Sarkar, 2020), reducing the likelihood of substantial longer-term reductions in oil palm expansion by large operators. These price changes may nevertheless mean that alternative economic and livelihood options may compete more favourably economically with palm oil in tropical peatland areas (cf. Wich et al., 2011), possibly encouraging their uptake. However, evidence of land holding in tropical peatlands strongly suggests that financial problems linked to poor revenues and limited capital may contribute to plot abandonment (Yusoff, Muharam & Khairunniza-Bejo, 2017).

It is likely that many employed and casual workers in tropical peatland cities will lose their immediate income sources owing to lockdowns and other disruptions, leading to migration of people from cities to home communities, with media reports suggesting this may already be happening in Indonesia (Listori, 2020), Malaysia (Radhi, 2020) and Peru (Collyns, 2020a; E. N. Honorio Coronado, 2020, personal observation). There is concern that this ‘returning home’ may bring sources of infection to vulnerable communities, leading to media reports of the strict closure of some indigenous territories (Sierra Praeli, 2020). This situation may impose a triple strain on tropical peatland communities by increasing the risk of infection via returning community members, imposing additional burdens on household resources and depriving households of external income sources. Furthermore, during previous periods of economic instability in Indonesia, such as following the 1998 financial crash, illegal land uses increased, including illegal logging (EIA-TELAPAK, 1999), as people sought alternative economic means. Economic disruptions are also expected in communities dependent upon eco-tourism (Evans et al., 2020; Hockings et al., 2020; Lindsey et al., 2020). There is thus a risk that the financial burden caused by the COVID-19 pandemic could increase unlawful exploitation of natural resources (timber, wild animals), as well as increasing other marginal livelihood practices (e.g. artisanal mining).

Food security

The UN World Food Programme has predicted that the COVID-19 pandemic will lead to over a quarter of a billion people suffering acute hunger by the end of 2020, owing to increased conflict, reduced aid and trade, price fluctuations and lost incomes (Anthem, 2020). Concerns have been raised that the pandemic is affecting all four pillars of food security (food availability, utilisation/nutrient intake, stability and particularly food access) and is leading to forced cut-backs in nutrient-rich non-staple foods towards starchy staples, with consequent long-term adverse health impacts (Laborde et al., 2020). We have already suggested that, given existing relatively high poverty levels and frequent lack of food self sufficiency, plus positions at the end of long supply chains for many external commodities, tropical peatland communities may be particularly vulnerable to food supply problems and price rises. Furthermore, while statutory and NGO food assistance programmes are operating and good transport links exist in some tropical peatland areas, these are designed to deal with relatively small numbers of clients and could be quickly overwhelmed if demand increases dramatically, whilst other areas have poor assistance infrastructure and poor, or often no, road access (e.g. Fig. 3). For indigenous communities, such issues may be exacerbated by a common lack of visibility in public policy (United Nations, 2009). Ensuring food security and the provision of food assistance to tropical peatland areas thus presents logistical, organisational and political challenges, and development of more localised food supply chains may therefore be favourable in post-pandemic responses (as recommended in a general context by Pearson et al., 2020).

As previously mentioned, tropical peatland community livelihood strategies are often based on a range of economic activities, whose relative importance varies temporally. For example rural communities in Borneo have for centuries been fairly entrepreneurial and responsive to changing global/regional trade patterns (Arenz et al., 2017; Dove, 2011). From their perspective, the COVID-19 pandemic may therefore not be viewed exceptionally, but rather as yet another development to which they must respond. Consequently, any potential loss of economic opportunities and reduced external demand for, or price of, locally produced commodities in such communities may lead to individuals reconsidering their livelihood strategies, resulting in changes in natural resource exploitation. For example any price declines in swiftlet nests produced in Indonesian peatland areas following potential economic slowdowns in China (the main market), may lead swiftlet keepers to search for alternative incomes from harvesting forest resources. While such shifts may not necessarily result in increased pressure on natural resources, concern nevertheless appears warranted, as recent reviews suggest that increased overall local environmental resource pressures may be expected under conditions of rural economic hardship (Robinson, 2016) and that, where poverty exists, households are more likely to pursue economic activities that improve their family’s short-term livelihoods, regardless of environmental impacts (South-east Asia: Douglas, 2006; Peru: Schulz et al., 2019b). From a positive perspective, any such changes may present an opportunity to work productively with affected communities to innovate and find more sustainable, locally rooted ways of responding to their needs.

Nevertheless, research suggests that experience of adverse shocks leads to increased risk aversion (Gloede, Menkhoff & Waibel, 2015) and that communities may be less willing to engage in alternative ‘sustainable’ livelihood activities if these are perceived to carry increased risks (Rodriguez et al., 2009). Perceived threats to health, income and food security related to COVID-19 may therefore reduce uptake or outcomes of revitalisation and community development activities in tropical peatland areas. Indeed, work in Indonesia has demonstrated that ecosystem restoration activities are unlikely to be adopted unless accompanied by assurances concerning food and income security (Carmenta & Vira, 2018).

The post-(peak) pandemic pressures governments face to ‘rescue the economy,’ may lead governments and companies in tropical peatland areas to reduce (or at least withhold attempting to improve) social and environmental standards (e.g. media reports: Carrington, 2020; Hurowitz, 2020), in an effort to reduce commodity prices and thus increase sales to rescue or expand industries such as palm oil. While evidence is currently limited, some media reports indicate that the pandemic may be employed to chip away at existing environmental measures, such as the EU’s palm oil ban (Kobo, 2020). Such trends could be exacerbated if economic restrictions and priority shifts lead to reduced enforcement of environmental and social regulations concerning extractive industries, or reduced emphasis on conservation and restoration, in tropical peatland areas.

Land conflicts

Land conflicts are one of the most important and complicated problems in land-use management in many countries with large peatland areas (Colchester, 2010; Colchester et al., 2007; Nesadurai, 2013; Scullion et al., 2014). In Peru, 24% of the estimated peatlands of the Pastaza-Marañon overlap with oil-extraction or exploration concessions, national reserves and territories of indigenous communities (Roucoux et al., 2017). In Indonesia, 807,178 ha are in land tenure conflict, with 73% of this contested land in the plantation sector (KPA, 2019). Increasing economic pressures and food insecurity associated with the COVID-19 pandemic may potentially aggravate this situation, with any increases in land tenure conflict likely to lead to increased peat fire incidence, given previously reported links (Medrilzam et al., 2014; Suyanto, 2007), and potentially therefore to further disenfranchisement and displacement of the poorest households and communities. This, in turn, may trigger further habitat encroachment and increased human-wildlife contact, whilst both food insecurity and haze pollution in degraded tropical peatland areas may increase COVID-19 impacts and susceptibility, as we have previously outlined. A glimpse of the complex and uneven effects that can result from such large-scale disruptions is found in Schreer’s (2016) ethnography of a Central Kalimantan village’s experiences and aspirations in the early- to mid-2010s. Here, the end of the logging boom—precipitated by rapid governmental decentralisation—in the mid-2000s had highly variable effects in her fieldsite, with migrants and non-migrants reacting differently, and other industries and initiatives bringing in new opportunities, risks and land conflicts, as well as widespread disillusionment and resentment at various external parties (Schreer, 2016).

The return migration of casual and temporary workers from cities in to rural tropical peatland communities and the continued presence of migrant labour (working in industries disrupted by COVID-19), as noted above, constitutes a further factor that may exacerbate pressure on land resources and thus increase conflicts, as well as acting as a direct vector for disease transmission. Supporting local communities in and around tropical peatland areas to resolve land tenure conflicts and promote inclusive, sustainable community-led management will likely prove important in mitigating these risks surrounding land conflict issues.

Unequal community and gender impacts

Importantly, any negative economic, social and health impacts for communities will not occur in a vacuum, and the social and ecological impacts of COVID-19 will affect different communities and community members unequally. Regional inequalities, and differences in access to health facilities between rural and urban areas, are likely to impact (efforts to curb) COVID-19 impacts. For example, in Indonesia, COVID-19 testing rates in Java have only surpassed WHO minimum recommendations in the capital, Jakarta (WHO, 2020b), and other provinces with large peatland areas generally lag behind in terms of health facility availability and access (MoH RoI, 2019). Furthermore, there is evidence that legal and policy frameworks support development patterns that marginalise peatland communities, resulting in the neglect of their interests by policy makers (McCarthy & Robinson, 2016). The COVID-19 crisis may exacerbate these trends, contributing to increased levels of economic (Payne & Bradley, 2020) and intersectional inequality (Bowleg, 2020). It is being increasingly recognised that tropical peatland communities are not homogenous in relation to economic status, ethnicity, access to facilities, etc. (Thornton, 2017) and the interaction of these factors means that the impact of COVID-19 is therefore highly unlikely to affect all members of tropical peatland communities equally, with impacts likely varying between households.

In addition, studies have highlighted how economic, food and health crises often put women in a more vulnerable position compared to men, despite shrinking options available for both genders (Pitkin & Bedoya, 1997; Pollock & Lin Aung, 2010). For example previous studies on other diseases such as tuberculosis have demonstrated context-specific gender-related differences in the barriers to access diagnostic and treatment services, especially in developing countries (Krishnan et al., 2014), while the greater role of women as primary care givers may increase their probability of infection (Davies & Bennet, 2016; Wenham, Smith & Morgan, 2020). In line with this, international organisations have already highlighted the key role that women play in food provision in indigenous rural communities and the likely heightening of this and other burdens on women, including the loss of childcare and other support services, plus surges in domestic violence, during the COVID-19 pandemic (FAO, 2020; United Nations, 2020; UN/DESA, 2020). For example the closure of schools is likely to disproportionately increase the childcare burden on mothers and older daughters within families (De Paz et al., 2020). Recognising and accounting for such gender differences will be important for mitigating the impact of COVID-19 in tropical peatland areas. Related to this, care should also be taken in pursuing novel economic developments, such as the expansion of independent oil palm smallholdings, which may risk exacerbating or even producing new gender inequalities (Elmhirst et al., 2017; Julia & White, 2012) that may in turn compound gender inequalities relating to COVID-19.

Research, training and education

The COVID-19 pandemic is significantly impacting field research globally, with travel restrictions and social distancing, plus (potential) reductions in research funding, leading to the adjustment, postponement or cancellation of many ongoing and planned field activities (Corlett et al., 2020; Evans et al., 2020; Lindsey et al., 2020). Among other impacts, research disruptions could potentially delay or prevent key output production (with possible knock-on effects on policy development), cause data gaps (particularly for long-term ecological data collection) and equipment supply issues. Social research, especially that involving bringing people together physically, will be particularly impacted (Corlett et al., 2020). For example, at the time of writing, the Indonesian Ministry of Research and Technology has prohibited new foreign researchers from entering and conducting research activities until the pandemic has ended (RISTEK-BRIN, 2020). In DRC, the UKRI-funded CongoPeat project is continuing to operate virtually but field activities and face-to-face initiatives and meetings have been paused, with field researchers required to depart suddenly and leave collected samples behind, thus delaying or potentially even risking completion of their analysis (C. E. N. Ewango, G. Dargie & D. Kopansky, 2020, personal observation). Media reports indicate that some indigenous communities have effectively closed off to visitors to protect themselves from the potentially devastating impacts of COVID-19 on their communities (Peruvian Amazon, including the peatland region of Loreto: Sierra Praeli, 2020; Papua: Milko, 2020). Similar observations have also been made for remote villages in Central and West Kalimantan, Sabah and Sarawak (L. Chua & M. A. Imron, 2020, personal observation), with such measures representing a traditional mechanism among rural Borneo communities for protecting themselves from the spiritual and other effects of dangerous events like deaths or disease outbreaks (Chua, 2012). Even after the COVID-19 pandemic (peak), such communities may retain their wariness, remaining less welcoming or even hostile to outside researchers, who have an ethical obligation to ensure that they do not inadvertently transmit the disease to these communities. Research on great apes and other primates is also at high risk of disruption, as measures are put in place to prevent reverse zoonotic transmission of SARS-CoV-2, given the perceived vulnerability of these species (Damas et al., 2020; Gillespie et al., 2020; IUCN SSC WHSG & PSG SGA, 2020; Melin et al., 2020). This has led some to question whether all primate field research should be cancelled for 2020 (Reid, 2020), with others highlighting the potential negative impacts of such a move and the potential primate conservation and research opportunities that may arise from the pandemic (Lappan et al., 2020; Trivedy, 2020). Ultimately, the risks of any research in relation to the COVID-19 threat will need to be carefully balanced against the counter-risks of not conducting or restricting research in terms of addressing other important conservation and community wellbeing issues. From a positive perspective, some of these developments may facilitate the empowerment of local scientists and thus help strengthen in-country scientific research in tropical peatland areas.

Research and training meetings are being cancelled or postponed around the globe, including rescheduling of the International Peatland Congress from June 2020 to May 2021, which will negatively impact both networking opportunities for peat scientists and short-term income for the International Peatland Society. Disruptions to teaching, research and networking activities, and related publications, will potentially have reverberating impacts on the careers and (short-term) incomes of students and junior researchers, with possible overall negative impacts on future scientific expertise (Corlett et al., 2020). For example, Borneo Nature Foundation (BNF) and Universitas Gadjah Mada (UGM) had been developing plans for BNF to support selected UGM students to conduct research dissertations on tropical peatlands in 2020. These have now been converted to desk-based projects, which cannot provide students with vital first-hand perspectives of the tropical peatland environment. More widely, impacts are likely to be especially severe and potentially irreversible for students and researchers already facing (economic) disadvantages, acting to further entrench existing biases and low representation of black, Asian and minority ethnic (BAME) groups in environmental organisations (Taylor, 2015), and women in tropical peatland research (Thornton et al., 2019). Parents, and particularly mothers, are also likely to be disproportionately affected (Staniscuaski et al., 2020). Some positive responses are, however, already evident in tropical peatland nations, with for example UGM instituting a series of online expert talks, BNF conducting live children’s education sessions through Facebook Live (Borneo Nature Foundation (BNF), 2020a) and some schools in remote areas of Kalimantan without internet using radio broadcasts to deliver teaching during the pandemic (Jakarta Post, 2020).

There is also the risk that research and development funds will be redirected towards COVID-19-related projects, increasing the difficulty of accessing funding for other research and conservation priorities (Evans et al., 2020). Funding shortfalls for tropical peatland research may lead to local research and other staff temporarily losing income tied to specific projects, or even redundancies of permanent staff. While some international NGOs may be able to furlough some staff to receive government funding support, this is unlikely to be an option in most tropical peatland countries. In addition to negatively impacting local economies (tropical peatland research is often conducted in remote rural areas with limited economic opportunities), redundancies of permanent research staff would also represent a consequent loss in local research project capacity if staff move to work in other sectors, which may be difficult and require a long time to replace. On the plus side, prolonged international travel restrictions may help promote the role of local researchers in multi-national research projects, while also reducing the carbon footprint of research involving international flights. Whilst it may be possible to maintain communications and coordination of many ongoing projects using online communication tools, this may be limited by lack of or slow internet connections in more remote tropical peatland areas. Equipping remote communities with such communication infrastructure would help minimise this and other wider impacts resulting from reduced potential for outsider travel to these areas. While researchers may be able to help contribute some individual-based assistance (e.g. laptop provision, booster aerials), provided sufficient flexibility exists in research grant provisions, larger-scale upgrades in communications infrastructure would require in-country government support. Potential may also exist for international travel funds to be re-allocated to support online learning or other training for local students and researchers, helping to build local capacity in the long term, although in the shorter-term international travel to undertake studentships and receive training may be curtailed.

Conservation and restoration

In addition to potential funding impacts discussed in the previous section, field activity disruptions, delays and cancelations will likely detrimentally impact many tropical peatland conservation, restoration, community development and outreach projects (e.g. Indonesia: Gan et al., 2020). This is particularly so when activities are time sensitive (e.g. planting seedlings during periods of optimal peat water level), require continued maintenance (e.g. tending seedlings in nurseries that may otherwise die, requiring much time and funds to replace), or typically involve large teams or in-person gatherings. For example, BNF temporarily halted all of its in-person childrens’ education and other activities involving groups of people, including community development, sustainable livelihood and fire-fighting training (Borneo Nature Foundation (BNF), 2020b). Some organisations have initiated self-isolation protocols before and after field visits as a COVID-19 precaution (e.g. Frankfurt Zoological Society in Jambi, Sumatra: T. Lestarisa, 2020, personal communication), which although potentially effective in relation to reducing transmission risk, may place additional demands and stresses upon staff.

Halting dialogue with tropical peatland communities, who are frequently remote and have limited or no internet access, may lead to reduced uptake or failure of alternative livelihood development activities, if critical implementation periods are missed (e.g. a planting season), enthusiasm wanes, community members seek alternative opportunities and become tied to these, or if initiatives are incorrectly implemented and so fail because of reduced training provision, thus potentially decreasing local enthusiasm for such initiatives in future. Even activities that can be postponed may still suffer from negative impacts, if this means that calendar-based targets are not met, community engagement drops, or consequent additional damage occurs (e.g. dams to restore peatland hydrology are damaged owing to lack of project presence in the area). As noted in the previous section, equipping rural communities with improved communications infrastructure would help minimise such impacts.

As is the case more generally (Corlett et al., 2020; Evans et al., 2020; Hockings et al., 2020; Lindsey et al., 2020), TPSF encroachment, timber and wildlife harvesting is likely to increase if conservation agencies are less active or visible, or enforcement reduced, particularly if rural communities do increasingly fall back on exploiting natural resources owing to economic shock or food insecurity (see above). For example, media reports quoting conservation organisations have already indicated increased poaching in the Leuser Ecosystem, Sumatra, which includes TPSF areas (Hanafiah, 2020). Importantly, if work restrictions or funding shortfalls lead to reduced fire-fighting capabilities in degraded tropical peatland areas, this may result in increased fire incidence and severity, as highlighted by reputed researchers and institutions in media reports (Cannon, 2020; Taylor, 2020). Fire-fighting in South-east Asian peatlands is often undertaken by local community members or, in plantations, by company teams. In these situations, Moore et al. (2020) identify COVID-19 infection risks either from a home setting to fire teams and, given the difficulties in social distancing during fire-fighting, from the fire team back to the home and community. This could impact on the effectiveness of fire-fighting, as well as on individuals and their families. As noted above, any resulting increases in TPSF degradation, fragmentation, wildlife harvesting, or haze pollution will be expected to increase the immediate impacts of the COVID-19 pandemic and/or future potential for zoonotic EID emergence in tropical peatland areas.

As a result of the above, plus the potential for SARS-CoV-2 transmission to susceptible wildlife species, there is a risk of significant negative impacts on tropical peatland biodiversity. This is likely to be particularly important for peatland specialists and already threatened species. This includes, for example the orangutan (Pongo spp.): the two species occurring on tropical peatland are Critically Endangered, with low and rapidly declining populations, have limited distribution on only one island each (Sumatra: P. abelii, Lesson; Borneo: P. pygmaeus, Linn.), have significant proportions of their populations in tropical peatland, are unable to persist in completely deforested landscapes and are at high risk from hunting owing to their slow reproductive rates (Utami-Atmoko et al., 2017). Orangutans and other non-human primates (Damas et al., 2020; Melin et al., 2020) are also potentially at risk of spill-over of the virus, including from asymptomatic human carriers, leading to recommendations for ape-based eco-tourism, field research and non-essential habitat conservation activities to be reduced, and impacting the activities of rescue, rehabilitation and release organisations (Borneo Orangutan Survival Foundation (BOSF), 2020; Gillespie et al., 2020; IUCN SSC PSG SGA, 2020; IUCN SSC WHSG & PSG SGA, 2020; Reid, 2020). The crisis thus increases the urgency to fully implement the IUCN best practice guidelines on ape tourism and health monitoring/disease control (Gillespie et al., 2020; IUCN SSC WHSG & PSG SGA, 2020). While eco-tourism is not currently commonplace in tropical peatlands, disruptions to ape-based eco-tourism may potentially threaten the viability of ape/habitat conservation and livelihood initiatives in those tropical peatland areas relying heavily on this source of income, and result in shifts back towards more destructive economic activities (cf. Evans et al., 2020; Hockings et al., 2020).

How might COVID-19 impact future tropical peatland conservation and what repercussions might this have in relation to disease pandemics?

Recent comprehensive reviews have highlighted the multiple, inter-linked threats and challenges already facing tropical peatland conservation and restoration, and provided suggestions to help tackle these (Dargie et al., 2019; Harrison et al., 2020; Roucoux et al., 2017). The additional short- to medium-term considerations arising in the context of the COVID-19 pandemic that we outline above may not all come to light, may create only short-term issues if the pandemic proves to be relatively short lived and recovery is relatively rapid, or may create longer-term changes if the pandemic is long-lasting or creates irreversible shifts. The severity and duration of impacts experienced will inevitably also vary between and within tropical peatland nations. We cannot therefore provide firm predictions regarding longer-term impacts of the pandemic on tropical peatlands and their communities, and consequently on changes in the potential of tropical peatlands to act as a source of future zoonotic EID pandemics. This caveat notwithstanding, we highlight some potential longer-term changes that may be important for policy makers, scientists and other tropical peatland stakeholders to consider going forward.

Some tropical peatland nations have already indicated that development projects will be paused or cancelled, and that economic growth may be prioritised above sustainability concerns in post-pandemic recovery phases, which could have long-term repercussions. For example, media reports indicate that Indonesia’s planned capital city relocation may be postponed owing to the pandemic (Demetriadi, 2020; Gokkon, 2020), whereas Padat Karya (Labour Intensive) government projects involving large numbers of local community members are still proceeding and being prioritised as an attempt to mitigate the economic impacts of the crisis (Iskandar, 2020; Widodo, 2020), as are 89 major infrastructure development projects, including potential development of large peatland areas for food production in Central Kalimantan with the goal to improve food security (Jong, 2020a). A history of prioritising economic growth over public health in the context of haze from peatland fires has been suggested in Malaysia (Varkkey & Copeland, 2020), while some recent media reports indicate that Indonesia will prioritise economic growth following the pandemic, with the consequence that emission reduction targets will not be increased (Jong, 2020c) and environmental protections potentially rolled back (Jong, 2020d). In Peru, the largest threat is the (re-)activation of regional economies to promote agricultural development (Peruano, 2020), which with the current lack of any documents to guide land use planning, may increase deforestation and access to remote forests (e.g. potential road construction that will affect peatlands in Loreto: Baker et al., 2020).

The race to return to economic growth and stability could also drive increased fossil fuel extraction. This is a risk in the Congo Basin (Dargie et al., 2019; Miles et al., 2017) and Peruvian Amazon (Lilleskov et al., 2019), where many oil concessions coincide with peatland areas. In Peru, oil extraction and transportation has led to many spills from poorly maintained pipes that have often been slow to be remediated (Rodríguez Mega, 2016). Expanding oil exploration and bringing new concessions into production involves infrastructure creation, clearing seismic survey lines, and building access roads and pipelines (Roucoux et al., 2017), typically causing habitat loss and degradation (Laurance, Goosem & Laurance, 2009; Mäki, Kalliola & Vuorinen, 2001). We note, however, that conversely and at least in the short-term, an alternative outcome of the pandemic could be lower global oil demand and production through at least 2020 (OGJ Editors, 2020), which may result in oil exploration becoming a lower immediate priority. Indeed, it has been indicated by the Ministry of Environment and Tourism that oil exploration in RoC peatland areas will not proceed in accordance with the country’s international climate commitments (A. Soudan-Nonault, 2020, personal communication; see also Weston, 2020).

Concerns have been raised regarding the intersection of the COVID-19 outbreak and public health responses with the ongoing increase in climate hazards associated with climate change (Phillips et al., 2020). Tropical peatlands are vulnerable to climate change, which will likely result in increased drought frequency, duration and/or severity, with consequent peat water table lowering, and increased peat oxidation and fire incidence, as already seen extensively in South-east Asia (Turetsky et al., 2015). There thus exists a risk of a positive-feedback loop development (cf. Harrison et al., 2020), whereby any decisions or events related to the pandemic that lead to increased peatland degradation and fire also then lead to further carbon emissions and peatland degradation, thus increasing future fire prevalence and the various negative impacts associated with this, including in relation to COVID-19 and potential future disease pandemics.

While peatland rewetting and revegetation may help mitigate some impacts of potential increases in the areas of degraded and burned peatland, these processes are unavoidably slow in nature, with peat burned in a single fire event taking potentially hundreds of years to re-form, generating concerns regarding the political tractability of some tropical peatland restoration goals (Harrison et al., 2020). Coupled with the existing large areas of degraded peatland already requiring restoration, particularly in South-east Asia, any potential increase in the extent of degraded and burned peatland as an indirect consequence of the COVID-19 pandemic would be expected to have long-term negative repercussions for tropical peatland management, community livelihoods, public health, carbon emissions and biodiversity. Reversing such changes, and enhancing the sustainability of tropical peatland management in general, may also become more challenging in the foreseeable future if, as discussed above, the current pandemic leads to loss of skilled, in-country peatland scientists and conservationists who would take time to replace. Any future pandemics may obviously exacerbate this.

Conclusions and Recommendations

Our review indicates that sustainable management and conservation of tropical peatlands is important for mitigating the impacts of the current COVID-19 pandemic, while avoiding further encroachment into these ecosystems and intensive harvesting of their wildlife could reduce the potential for future zoonotic EID emergence and severity (summarised in Fig. 1). Importantly, because many of the potential impacts identified in relation to COVID-19 arise through socio-economic disruptions relating to our response to the pandemic, rather than unique characteristics of the SARS-CoV-2 virus itself, this conclusion should be applicable more widely to pandemic situations and similar sudden socio-economic shocks. To our knowledge, this link between tropical peatland ecosystems and zoonotic EIDs has not been specifically noted previously and this therefore represents an important finding in terms of our understanding of the benefits that these ecosystems provide, thus strengthening the argument for their conservation and sustainable use. In addition, and recognising that our assessment is both preliminary and likely incomplete owing to the recent emergence and rapid evolution of the COVID-19 pandemic, we highlight the potential additional threats and challenges towards sustainable management of tropical peatlands and their wildlife posed by this pandemic, and the potential knock-on effects of these for tropical peatland conservation and local communities going forward.

These conclusions support and strengthen previous recommendations regarding the sustainable management of tropical peatlands, including regarding hydrological functions, fire prevention, avoiding encroachment and habitat fragmentation, plus maintaining and where necessary restoring healthy peatland ecosystems that support diverse biological communities, sequester carbon and provide socio-economic benefits to local communities (see, for example Dargie et al., 2019; Dohong, Abdul Aziz & Dargusch, 2018; Graham, Giesen & Page, 2017; Harrison et al., 2020; Mizuno, Fujita & Kawai, 2016; Roucoux et al., 2017; Schulz et al., 2019a; Schulz et al., 2019b; Wijedasa et al., 2018 and references therein for detailed recommendations). They also support previous recommendations to carefully manage terrestrial wildlife harvesting in and trade from tropical peatland areas (Bodmer & Lozano, 2001; Harrison et al., 2011) and more generally (Corlett, 2007; Harrison et al., 2016). Further to these generic recommendations, we offer the following specific suggestions for researchers and practitioners relating to tropical peatland sustainable management, COVID-19 and potential future zoonotic EIDs:Field projects should support frontline team members to reduce their COVID-19 exposure risk and, thus, the risk of them infecting other community members and of work being impacted through worker infection. In addition to ensuring and facilitating adherence to current government and WHO guidelines, specific recommendations have been proposed for fire-fighters (Moore et al., 2020), and work with great apes (IUCN SSC WHSG & PSG SGA, 2020) and in their habitats (IUCN SSC PSG SGA, 2020). Development of similar specific guidelines for other sectors relating to tropical peatland management could be beneficial.

All actors should support local communities in and around tropical peatland areas to resolve land tenure conflicts and promote sustainable community-led management (Blackman et al., 2017; Santika et al., 2017), facilitate locally-derived sustainable economy development (Pearson et al., 2020), and prevent external users from unsustainably exploiting these ecosystems. This includes encouraging, and where necessary opening up new, discussions regarding the relationships between use of and encroachment into tropical peatlands, commercial vs. subsistence wildlife harvesting, and zoonotic EID risk, and considering these holistically in the context of community wellbeing and aspirations (cf. Kavousi et al., 2020). It should also include re-orientating discussions, from conservation and development organisations approaching with a perspective of ‘how can we help them’ to an approach of ‘we’re all in this together, so what can we each bring to the table to help?.’ Such re-orientations should ideally be informed by fresh social research on how communities’ lives, relations and experiences have been impacted by COVID-19, as these are unlikely to simply revert to a pre-pandemic ‘normal’.

The peatland research community should rethink approaches to research in at-risk communities, to potentially incorporate more remote and online working, plus enhanced roles for local research teams. This may require re-appraising how likely increasingly limited research resources are distributed, and particularly supporting local research capacity development and local researcher empowerment within international collaborations. Incorporating and centering local knowledge and priorities from the outset has the additional benefit of increasing the success of conservation and development interventions (Mistry & Berardi, 2016). COVID-19 has forced increased use of online communications tools and we hope this can help foster new patterns of international collaboration that further empower local researchers. See Lupton (2020) for an extensive list of potential approaches to reducing COVID-19 risk during social science fieldwork.

Wherever possible and working within all relevant government and sector COVID-19 guidelines, research, conservation and community projects should continue collection of long-term monitoring data and initiate repeat data collection to provide before-and-after comparisons, in order to help assess the impacts of the COVID-19 pandemic in relation to the issues discussed in this paper. This may include, for example, patrol team or satellite data monitoring to assess if forest incursions or fire incidence have increased, or community livelihood data to assess if livelihood choices have changed and if/how this has impacted local wellbeing. It should also include collecting baseline human and wildlife community health information to better understand, monitor and thus mitigate zoonotic EID risk in tropical peatland areas.

Policy makers in tropical peatland areas should ensure that their policy decisions do not amplify the risks of the COVID-19 or potential future pandemics. In particular, any weakening of environmental and social standards and regulations relating to tropical peatlands should be avoided (see also Evers et al., 2017; Wijedasa et al., 2017). Instead, we recommend development and implementation of a ‘One Health/Ecohealth’ approach in tropical peatland nations, which recognises and supports the complex, mutually-beneficial interconnections among the health of people, animals, plants and our shared environment (El Zowalaty & Järhult, 2020; Hockings et al., 2020; Roger et al., 2016; UNEP & ILRI, 2020). We further recommend strengthening communications infrastructure in (remote) tropical peatland areas, to help minimise any impacts of reduced travel or on-the-ground activities in these areas.

Funders should exercise maximum possible flexibility to tropical peatland research, conservation and community projects impacted by the COVID-19 pandemic, including regarding achieving deliverable targets and deadlines, where possible offering additional support to maintain local staff employment during periods of enforced down time instead of offering zero-cost funding extensions, and not penalising unavoidable “failures” by current projects in future funding application rounds. This will be important in minimising potential project closures, plus staff redundancies or lost income, and associated risk of loss of expertise. Opportunities to diversify revenue sources to increase the resilience of conservation efforts should also be explored (Lindsey et al., 2020).

All actors should recognise the role that tropical peatlands, their conservation, sustainable management and restoration will play in both the current COVID-19 pandemic and in the potential for future zoonotic EID emergence; and actively promote this message in fund-raising, education, outreach, community and government engagement, while taking care to tailor messages and associated recommendations appropriately to local audiences (Chua, 2020; Kavousi et al., 2020; MacFarlane & Rocha, 2020; see also Charania & Tsuji, 2012; Chua et al., 2020).

As is the case more generally (Corlett et al., 2020; Evans et al., 2020; Hockings et al., 2020; UN/DESA, 2020; UN/EMRIP, 2020; UNEP & ILRI, 2020), our review indicates that the relationships and reciprocal impacts between tropical peatland ecosystems and communities, COVID-19 and disease pandemics are inter-linked, multi-faceted, and likely to vary over both space and time. This adds an extra pandemic-related dimension to the increasingly complex picture that is emerging regarding the challenges and opportunities for conservation and sustainable management of tropical peatlands (Dargie et al., 2019; Harrison et al., 2020; Roucoux et al., 2017). At the same time, this also suggests that potential for win–win solutions exists to simultaneously address the challenges identified herein relating to COVID-19 and disease pandemics in tropical peatland areas, alongside the previously identified challenges facing these ecosystems and their resident human communities. This is particularly pertinent in light of recent analyses estimating that global pandemic prevention costs associated with reducing deforestation, wildlife trade and farmed animal spill-over, and early zoonotic disease detection and control over the course of ten years is equivalent to only about 2% of the costs of the COVID-19 pandemic (Dobson et al., 2020). Moving beyond these conclusions linked to our review, and while we hope that the COVID-19 pandemic is rapidly mitigated and thus many of the potential issues discussed in this paper fail to (fully) materialise, we nevertheless trust that our consideration of these issues and recommendations provided helps improve our ability to anticipate and prevent potential negative impacts that may arise from the pandemic. In addition, and regardless of the pandemic, we contend that it helps to foster the inter-connected thinking that will be required to ensure the future health and wellbeing of tropical peatlands and their human communities alike.

Supplemental Information

Supplemental Information 1 Structured literature review approach.

Click here for additional data file.

Supplemental Information 2 Structured literature search results.

Click here for additional data file.

We thank our respective institutions for their support, Stuart Smith and Nur Estya Binte Rahman for discussions regarding literature reviews, and the many colleagues and tropical peatland communities that have engaged in fruitful discussions over the years and thus contributed towards our thinking in relation to the topics discussed in this paper. We are grateful to the Editor, Valéria Kaminski and one anonymous reviewer for constructive comments that helped improve the manuscript.

Additional Information and Declarations

Competing Interests

Author Contributions

Data Availability

Mark E. Harrison and Susan M. Cheyne are Directors of, and Sara Thornton is a researcher, with Borneo Nature Foundation International; Susan Page is a Trustee. Susan M. Cheyne is also Vice Chair of the IUCN SSC PSG Section on Small Apes. Lahiru S. Wijedasa is employed by ConservationLinks Pvt. Ltd. Dianna Kopansky and Johannes Refisch are employed by, and Julie Van Offelen is an independent consultant for, the United Nations Environment Programme.

Mark E. Harrison conceived and designed the experiments, performed the experiments, analyzed the data, prepared figures and/or tables, authored or reviewed drafts of the paper, and approved the final draft.

Lahiru S. Wijedasa conceived and designed the experiments, performed the experiments, analyzed the data, prepared figures and/or tables, authored or reviewed drafts of the paper, and approved the final draft.

Lydia E.S. Cole conceived and designed the experiments, performed the experiments, prepared figures and/or tables, authored or reviewed drafts of the paper, and approved the final draft.

Susan M. Cheyne conceived and designed the experiments, performed the experiments, authored or reviewed drafts of the paper, and approved the final draft.

Shofwan Al Banna Choiruzzad performed the experiments, authored or reviewed drafts of the paper, and approved the final draft.

Liana Chua performed the experiments, authored or reviewed drafts of the paper, and approved the final draft.

Greta C. Dargie performed the experiments, authored or reviewed drafts of the paper, and approved the final draft.

Corneille E.N. Ewango performed the experiments, authored or reviewed drafts of the paper, and approved the final draft.

Euridice N. Honorio Coronado performed the experiments, authored or reviewed drafts of the paper, and approved the final draft.

Suspense A. Ifo performed the experiments, authored or reviewed drafts of the paper, and approved the final draft.

Muhammad Ali Imron conceived and designed the experiments, performed the experiments, authored or reviewed drafts of the paper, and approved the final draft.

Dianna Kopansky performed the experiments, authored or reviewed drafts of the paper, and approved the final draft.

Trilianty Lestarisa performed the experiments, authored or reviewed drafts of the paper, and approved the final draft.

Patrick J. O’Reilly performed the experiments, authored or reviewed drafts of the paper, and approved the final draft.

Julie Van Offelen performed the experiments, authored or reviewed drafts of the paper, and approved the final draft.

Johannes Refisch performed the experiments, authored or reviewed drafts of the paper, and approved the final draft.

Katherine Roucoux performed the experiments, authored or reviewed drafts of the paper, and approved the final draft.

Jito Sugardjito performed the experiments, authored or reviewed drafts of the paper, and approved the final draft.

Sara A. Thornton performed the experiments, prepared figures and/or tables, authored or reviewed drafts of the paper, and approved the final draft.

Caroline Upton performed the experiments, authored or reviewed drafts of the paper, and approved the final draft.

Susan Page conceived and designed the experiments, performed the experiments, prepared figures and/or tables, authored or reviewed drafts of the paper, and approved the final draft.

The following information was supplied regarding data availability:

All information, search criteria and results, and literature sources cited are available in the article and the Supplemental Files.

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
