# Peer review of "Tropical peatlands and their conservation are important in the context of COVID-19 and potential future (zoonotic) disease pandemics"

_PeerJ, doi:10.7717/peerj.10283_

## Round 0.1 · original submission · Minor Revisions

Thank you for what is an interesting and timely paper that places the importance of tropical peatlands in the context of zoonotic disease. The reviewers have provided what I believe is useful feedback on your manuscript and I look forward to reading the corrections and seeing your rebuttal. One of the reviewers noted that the last paragraph is an opinion and not a summary as it is a literature review not an opinion piece. I agree with this, however I believe these opinions are important and would suggest providing a strong summary before moving to what is clearly an opinion e.g. saying "Moving beyond the findings of this literature review..." (or some similar wording) so that it is distinct and therefore disassociates what is the fact based literature review from the author's opinions.

Reviewer 1 ·

Basic reporting

- The topics covered in the article are of great interest in the current and future scenario;

- Abstract, line 63: I suggest: “many potential vertebrate and invertebrate vectors”;

- Introduction: The introduction is well structured and the objectives of the review are well defined;

- Figure 1: Please, remove the Earth illustration from the center of the virus. This is unnecessary. I also suggest connecting the green box “increased biodiversity loss” (cause) with the box “increased human-wildlife contact” (consequence);

- This is a cross-disciplinary review within the scope of the journal.

- The article is written in good English;

- The revised literature is updated.

Experimental design

- The methodology is interesting (it looks “sincere” and consistent) and adequately described. In brief, the methodology is well structured for a review article;

- Table S1: The authors state “non-zoonotic diseases (e.g. flu)”. Flu is not a good example since some authors consider flu a zoonotic disease (usually human influenza strains is derived from non-human animals). Please, change to another example to avoid confusion;

- Table S2: Okay;

- Sources are adequately cited and the text is organized logically.

Validity of the findings

- Although the article often has very opinionated parts, references were used appropriately;

- The text is drafted in an appropriate manner and respecting a logical structure;

- Figures 2-6: Okay;

- Lines 864-869 (conclusion section): I suggest removing or modifying it for a more "pragmatic" paragraph. This final paragraph is full of expressions like "we hope". The authors must not forget that this is not an opinion article, but a review article, and therefore the final paragraph should represent the main conclusions about the works that have been reviewed throughout the text.

Additional comments

- Regarding the topic “Are Tropical Peatlands a Potential Source Habitat for Disease Pandemics?”: I suggest to include a paragraph (maybe second paragraph, after line 200) discussing the risk of spillover events associated with human contact with bushmeat and wildlife in forest environments;

- Lines 285-286: Please, remove “It is therefore possible – and indeed we hope – that many of the concerns raised here will never be realised in the context of this pandemic.” This sentence is speculative;

- Line 843: Please, correct “Wijedasa & al., 2017”.

·

Basic reporting

This review presents broad and cross-disciplinary interests that fall within the scope of the journal. The authors present recent and very important issues that are indeed accessible to researchers of different areas of knowledge. Besides, the authors raised important and current concerns. The Introduction section is clear. However, I strongly suggest some recommendations to better align the Introduction with the text body with the proposal, such as outlined as follows:

# The present paper raised very important questions regarding the impacts of the current global scenario caused by Covid-19 pandemic. The authors indeed show the importance to keep the existing conservation strategies in peatland areas at this moment and elegantly expose the likely consequences of a halt in investments in training scientists, people education, and conservatory demands along with the need of economic improvements to avoid community crisis. However, the paper is not really focused on direct links between Covid-19 and tropical peatlands. Considering this, the conclusion and introduction should be constructed into the environmental health of peatland areas (and its communities' livelihoods) and their connected concerns in times of pandemics instead of a direct consequence of the pandemics or risk for EID. Of note, these last issues should appear as they are in the text, but the reader should be aware of the actual discussions raised through the text.

# Regarding the title, some adjustments could be make it more suitable to the text body. Recalling the comments above and the elegant issues raised mainly in the sections "Economy and Livelihoods", "Food Security and Land Conflicts", and "Unequal Community Impacts", I encourage the authors to reformulate the review title, raising a different call similar as "COVID-19 AND TROPICAL PEATLAND AREAS: LESSONS ABOUT VULNERABLE COMMUNITIES, CONSERVATION STRATEGIES, AND RESEARCH THREATS IN TIMES OF PANDEMIC".

Experimental design

The review content is within the Aims and Scope of the Journal and the authors did present adequate investigation with high technical and ethical standards. The methods used by the authors are descriptive and did present sufficient and detailed information to be confirmed or replicated (obviously considering the possible changes in the scenario outlined in the time of writing). The information provided did present a comprehensive and unbiased coverage along with an important and neglected issue in the context of the current global scenario. The references were adequately chosen and cited. Nevertheless, I have some recommendations regarding the paragraphs and subsections as follows:

# Within the topic "Public Health", from line 344 to line 350, there is a brief discussion regarding different pathogens and infections. This is an important and likely neglected issue, especially considering peatland communities and Sars-CoV-2. Considering this, it would be interesting to introduce a paragraph raising the possible co-infections and their complications to such communities in vulnerable areas. The paragraph should answer questions such as: Is there such a relevant co-infection risk in peatlands communities? How severe this risk would be? What are the most common infections that could represent complications for people affected by Covid-19? What are the strategies to manage such risks?

# In the "Public Health" section, from line 352, there is a discussion more related to environmental aspects than public health per se. I suggest including a topic named "Environmental Aspects" in which the text from lines 352 to 381 could be replaced. Alternatively, the authors could rename the topic to "Public Health and Environmental Aspects". Also, despite the great information provided in this section, it would be adequate to provide or briefly cite the alternatives to deal with the problems stated (similar to the strategies suggested for food security in lines 482-485 in the topic "Food Security and Land Conflicts").

# Considering that 'land conflicts are one of the most important and complicated problems in land-use management in many communities with large peatland areas, affecting a large number of people', this issue deserves a separate topic with a more profound discussion and evaluation. Besides, considering the specific interest of this review, the Covid-19 pandemic should be addressed in the context of land conflicts in a perspective of how it would make communities more vulnerable to the disease and, again, authors may present strategies to avoid or minimize such effects.

# Authors should be congratulated for including the topic "Unequal Community Impacts". Moreover, this is a very important and still neglected aspect of the impacts of Covid-19 pandemic in overall aspects. Raising this issue greatly increases the coverage and insight of the review work carried out, adding unprecedented value to the article and to the Journal. Of note, a more profound discussion about how women on peatland communities are differently affected by the pandemic should be included. Finally, I suggest change the section title to "Impacts of Unequal Communities and Gender Bias".

# The topic "Research, Training, and Education" is pretty welcome. Over-funding focused only on Covid-19 research is quite dangerous indeed and the authors elegantly provide examples of the consequences of short and long-term replacement of resources directed only to Sars-CoV-2 research in detriment of other scientific areas. To better finish this, it would be wonderful if authors suggest strategies to better assist the remote peatland areas with adequate technology such as internet support, remote censoring, and all the needs for the viability of future (without physical contact) researches and collaborations worldwide. Such strategies, once previously presented, could be recalled in line 646 of the topic "Conservation and Restoration".

# Line 695: "How Might COVID-19 Impact Future Tropical Peatland Conservation and Sustainable Management, and What Knock-on Effects Might This Have on the COVID-19 and Possible Future Pandemics?" is quite a long title. Please consider separating it in two 'title questions' and adequate the text to respond to each one of them, to better highlight each of the questions addressed. 

# Recommendations and Conclusion could be separated sections. Also, even presenting recommendations at the end of the article, I reinforce to include or cite the possible solutions just after a problem is presented in the text body.

# Line 826: online communications/ technological resources for distance communication are recalled. Please make it an additional item in the recommendations section.

Validity of the findings

This review presents an important perspective, which considers various logistical aspects of both research and access to vulnerable populations in peatland areas so that they have adequate assistance. Still, the economic and environmental aspects revisited, in parallel with social issues of paramount importance, raise issues that, although urgent, are still neglected. However, the issue of the risk of emergence of infectious diseases and a direct link with Covid-19 are aspects presented in a secondary way throughout the text, thus some changes must be made in the Introduction and Abstract of the article, as outlined previously. I reiterate that such changes do not make the article less relevant. It is a question of adequacy with the ideas outlined in the text that, indeed, in some points are affected by the pandemic or that raise the discussion about EID. Finally, the conclusion is adequate and quite complete and presents strategies for mitigating the problems presented.

Additional comments

Dear Author(s),

I must congratulate you on the initiative in writing a paper addressing the conservation of peatland areas and the impacts that these strategies may suffer in the current pandemic scenario. This is a very important perspective, which considers various logistical aspects of both research and access to vulnerable populations so that they have adequate assistance. Still, the economic and environmental aspects revisited, in parallel with social issues of paramount importance, raise issues that, although urgent, are still neglected. This is the overview I get from reading your manuscript. The risk of emergence of infectious diseases and a direct link with Covid-19 are aspects presented in a secondary way throughout the text, thus some changes must be made in the Introduction and Abstract of the article. I reiterate that such changes do not make the article less relevant. It is a question of adequacy with the ideas outlined in the text that, indeed, in some points are affected by the pandemic or that raise the discussion about EID.


# It sounds redundant when the authors use the term "emergence" and the abbreviation EID together. I suggest revising this throughout the text, with abstract as a starting point.

# Line 682: once transmission of Sars-CoV-2 from humans to other primates is mentioned, please use the term 'spillover'. Once the authors decide to not explain this term previously, it would be interesting to emphasize its meaning at this moment.

---

## Round 0.2 · accepted · Accept

Many thanks for submitting your revised manuscript. Having read the revisions as have the reviewers we are very happy with the corrections and recommend the paper be accepted.

Reviewer 1 ·

Basic reporting

No comments additional to those already presented in the first revision of the manuscript. The current revised version of the article is suitable for publication.

Experimental design

No comments additional to those already presented in the first revision of the manuscript. The current revised version of the article is suitable for publication.

Validity of the findings

No comments additional to those already presented in the first revision of the manuscript. The current revised version of the article is suitable for publication.

Additional comments

The authors responded to the reviewers appropriately. The modifications made were sufficient to allow the publication of the article.

·

Basic reporting

# After the modifications made by the authors, the title of the manuscript and the text body are now properly aligned.

# The modification made on the manuscript title had an excellent result, as well as it maintained the originality and identity of the group of authors.

# In view of the uncertainties related to possible complications in the cases of COVID-19
and occurrence of co-infections, the changes made in relation to this discussion are pretty adequate.

# In full agreement with the changes in “Public Health and Potential Combined Impacts from Haze Pollution” as well as the modifications in the new “Land Conflicts” sub-heading.

# The modification in the sub-heading “Unequal Community and Gender Impacts” deserves congratulations: it is a necessary topic with the potential to stimulate this same discussion in future articles from different areas.

# Changes made in "Conservation and Restoration" are sufficient.

# The shortened title of the sub-heading “How Might COVID-19 Impact Future Tropical Peatland Conservation, and What Repercussions Might This Have in Relation to Disease Pandemics?” is now much better and with more fluid reading.

# Thanks for adding more possible solutions to issues raised within the body text and for reinforcing the issue about online communications/ technological resources for distance communication in the recommendations section.

Experimental design

I have no additional or new declarations regarding the Study Design.

Validity of the findings

The author's responses are pretty convincing. I have no additional suggestions.

Additional comments

Thanks to the authors for accepting the small changes as well. Again, I congratulate the entire team for this beautiful review work and I thank the Editor and the Journal for the opportunity to contribute.